# Relationship between Food Allergy and Endotoxin Concentration and the Toleration Status at 2 Years: The Japan Environment and Children’s Study

**DOI:** 10.3390/nu15040968

**Published:** 2023-02-15

**Authors:** Makoto Irahara, Kiwako Yamamoto-Hanada, Miori Sato, Mayako Saito-Abe, Yumiko Miyaji, Limin Yang, Minaho Nishizato, Natsuhiko Kumasaka, Hidetoshi Mezawa, Yukihiro Ohya

**Affiliations:** 1Allergy Center, National Center for Child Health and Development, Tokyo 157-8535, Japan; 2Medical Support Center for the Japan Environment and Children’s Study, National Center for Child Health and Development, Tokyo 157-0074, Japan

**Keywords:** cow’s milk, egg white, endotoxin, food allergy, hen’s egg, IgE, IgG4

## Abstract

Changes in household endotoxin concentration may affect the prognosis of food allergy (FA), but data on the association between household endotoxin concentration and an already-developed FA are scarce. Thus, we investigated the association between environmental endotoxin exposure and tolerance to hen’s egg (HE) and cow’s milk (CM) using data from children participating in the Japan Environment and Children’s Study who had HE allergies (*n* = 204) and CM allergy (*n* = 72) in their first year of life. We grouped the endotoxin results into quartiles 1–4 (Q1–Q4). In children with HE allergy and with CM allergy, there was no significant difference in the prevalence of tolerance to HE and CM at 2 years old when comparing endotoxin levels of the children in Q1 with those in Q2, Q3, and Q4, respectively. However, subgroup analyses by the presence of eczema and causal foods revealed that children in Q1 had a lower prevalence of tolerance to foods in some subgroup analyses and lower causal allergen-specific immunoglobulin G4 levels. Although an individually based approach against endotoxin according to background characteristics, such as eczema and causal foods, is necessary, preventing excessive endotoxin removal might contribute to FA resolution in some children.

## 1. Introduction

An immediate food allergy (FA) is induced by allergen-specific immunoglobulin (Ig) E antibodies and provokes allergic symptoms. Consequent food eliminations lead to poor quality of life in children and their caregivers [1,2,3]. The prevalence of FA has dramatically increased [4]. In Japan, approximately 10% of children develop FA in their first year of life [5], with hen’s egg (HE) and cow’s milk (CM) being the major food allergens [5,6].

FA is a serious global issue [4,7]. Although several studies showed that 68–94% of children with FA against HE and more than 90% of children with FA against CM acquired tolerance after puberty, other articles showed only 50% of cases of children with FA against HE and CM resolve naturally by 2 years of age. [8,9,10,11,12,13]. In a part of children, it is difficult to archive early natural tolerance to HE and CM. Oral immunotherapy treatment for children with FA is beneficial and leads to an increase in the threshold dose, inducing symptoms by decreasing food allergen-specific IgE levels and increasing food allergens-specific IgG4 levels [14,15,16,17]. However, the modulating factors of oral immunotherapy are unclear.

The epithelial barrier hypothesis, which posits that various environmental exposure factors aggravate allergies by affecting the epithelial barrier, was recently proposed [18,19]. Endotoxin is a soluble lipopolysaccharide fragment derived from Gram-negative bacteria and exists in the environment [20,21]. The levels of atmospheric particulate matter with a diameter < 2.5 µm and of endotoxin in Japan are >10 times higher than those in Germany [22]. Although the association between endotoxin and airway inflammation, such as bronchial asthma, has been reported [21,23], the association between endotoxin and FA remains unclear. A cross-sectional study showed a positive association between exposure to higher levels of endotoxin and the prevalence of sensitization to food allergens [24]. However, the association between endotoxin and FA resolution in children who have already-developed FA has not been investigated.

Thus, we investigated the association between environmental endotoxin exposure and tolerance to HE and CM by examining allergen-specific IgE and IgG4 levels in children at 2 years old who had FA against HE and CM and IgE sensitization.

## 2. Materials and Methods

### 2.1. Study Design, Setting, and Participants

This study used data from a sub-cohort study of the Japan Environment and Children’s Study (JECS). The JECS cohort included 104,059 fetal records, and the participants were recruited between January 2011 and March 2014 from 15 regional centers in Japan [5,25,26,27,28]. The sub-cohort study included 5015 infants with more detailed information and was planned to last until the children were 13 years old. This study included children with an immediate allergic reaction after ingestion of HE or CM during their first year of life, and endotoxin concentration data were measured in a unified institute when the children were 18 months old (Figure 1).

The JECS protocol was reviewed and approved by the Ministry of the Environment’s Institutional Review Board on Epidemiological Studies (Institutional Review Board number: 100910001) and by the Ethics Committees of all participating institutions. The JECS was conducted in accordance with the principles of the Declaration of Helsinki and other national regulations and guidelines. Data for this study were used from the JECS datasets released in October 2019 (jecs-ta-20190930) and April 2021 (jecs-qa-20210401). Written informed consent to participate was obtained from the parents of all participants.

### 2.2. Variables

Information about the reaction and toleration status of HE or CM was obtained via a written questionnaire administered to the children’s caregivers. Children with detectable food-specific IgE levels to CM and HE and whose parents answered “Yes” to the question, “Have your children had any symptoms within 3 h after ingestion of breast milk, formula milk, or baby foods?” and answered, “Formula milk” or “Hen’s egg” when asked “What foods do you think provoked those reactions?” were classified as having HE and CM allergy in their first year of life. Children were classified as having tolerance to HE and CM at 2 years old if the parents selected “Our child usually eats this food.” regarding HE and CM. Children were classified as having an intolerance to HE or CM if the parents selected the following answers: “Our child has not eaten this food up to now,” “Our child partially eliminates this food now,” or “Our child has previously ingested this food, but now it has been completely eliminated.” Data regarding the study participants’ characteristics were obtained from their medical records and the written questionnaire administered to the children’s caregivers. Information regarding sex, siblings, birth weight, and mode of delivery was obtained from medical records after delivery. Maternal allergies and household income were confirmed using questionnaires administered during the second or third trimester of pregnancy. Other information obtained via questionnaires included the ownership of indoor pets when the child was 6 months old and whether the child was breastfed for the first year of life. We considered current eczema as eczema during the past year in children aged 2 years old and confirmed this using a partially modified, validated Japanese version of the International Study of Asthma and Allergies in Childhood (ISAAC) questionnaire in children aged 6–7 years [29,30,31].

### 2.3. Measurement of Endotoxin Concentration

When the children were 18 months old, JECS staff visited their homes to collect house dust, which contains endotoxin, from the children’s mattresses. The endotoxin concentration was calculated per 1 mg of house dust using a kinetic chromogenic *Limulus* amebocyte lysate assay (Kinetic–QCL; Lonza Japan, Sagamihara, Japan) [26].

### 2.4. Measurement of Food Allergen-Specific Ig Levels

Serum levels of egg white (EW)- and CM-specific IgE and IgG4 were measured when the study participants were 2 years old using a densely carboxylated protein microarray (AMERIC, Tokushima, Japan) [32,33,34,35]. EW- and CM-specific IgE levels measured using this method correlate with the allergen-specific IgE levels measured using the ImmunoCAP system (Phadia, Uppsala, Sweden). The measurement limit values of the food-specific Igs were 0.01 BUe/mL for IgE and 50.0 BUg4/mL for IgG4. Therefore, detectable HE- and CM-specific IgE levels were defined as ≥0.01 BUe/mL in this study [33]. HE- and CM-specific Ig levels were log-converted for use in statistical analyses.

### 2.5. Bias and Study Size

To minimize bias, we included all children who met the definition of an allergy to HE and CM. We did not predefine the sample size because this investigation was an exploratory and hypothesis-generating study.

### 2.6. Statistical Analysis

We confirmed the following characteristics of the children and then used Fisher exact test to calculate the *p*-values for gender, siblings (the presence), household income (more than or less than ¥4 million), maternal history of allergies (atopic dermatitis, allergic rhinitis, or bronchial asthma), the climate of residence, low birth weight (<2500 g), mode of delivery (Cesarean section or vaginal), indoor pets (dogs or cats), breastfeeding (even a little after the age of 1 year), and the presence of eczema during the past year (using the ISAAC questionnaire administered when the children were aged 2 years). Residential climates were classified according to the location of the regional center in the JECS using the Köppen climate classification system [36]. We evaluated the endotoxin concentrations per 1 mg of household endotoxin concentration when the children were 18 months old and classified the results into 4 quartiles (Q1, Q2, Q3, and Q4). To quantify the main outcome, we conducted logistic regression analyses of the association between the 4 endotoxin quartiles and toleration statuses of HE and CM when the children were 2 years old. The presence of eczema in both HE allergy and CM allergy was selected as a covariate based on the report of a significant difference [37]. Data on breastfeeding in children with HE allergy was also selected as a covariate based on the observation of significant differences by univariate analyses in characteristics.

We used Mann–Whitney U tests to compare EW- and CM-specific IgE and IgG4 levels when the children were 2 years old between children in the Q1 group and children in the Q2, Q3, and Q4 groups. Moreover, to confirm of association between EW- and CM-specific immunoglobulin levels and status of toleration statuses, we used Mann–Whitney U tests to compare EW- and CM-specific IgE and IgG4 levels when the children were 2 years old in toleration statuses of HE and CM. In the subgroup analyses of children with and without eczema at age 2 years, the prevalence of children who had tolerance to HE and CM was compared between children in the Q1 group and children in Q2, Q3, and Q4 groups individually by Fisher exact test due to concern about the interaction of the presence of eczema with endotoxin concentration. Also, in this subgroup, Mann–Whitney U tests were conducted to compare EW- and CM-specific Ig levels between children in the Q1 group of endotoxin concentrations and other quartile groups individually. Spearman rank correlation test was added to confirm associations between endotoxin concentration and EW- and CM-specific Ig levels in subgroup analyses.

All statistical analyses were performed using SPSS v24.0 software (IBM Corp., Armonk, NY, USA). *p*-values < 0.05 were considered significant.

## 3. Results

### 3.1. Participants

This study analyzed the data of 204 children with HE allergy and 72 children with CM allergy in their first year of life. Of them, 170 children (83.3%) had pruritis and urticaria in children with HE allergy, and 56 children (77.8%) had these symptoms in children with CM allergy (Appendix A). One hundred seven children (52.5%) and 38 children (52.8%) had a tolerance to HE or CM, respectively, at 2 years old. Also, at 2 years old, 70 (34.3%) children with HE allergy and 34 (52.1%) children with CM allergy had eczema during the past year (Table 1).

### 3.2. Endotoxin Concentrations and House Dust Mite Allergens

The four quartiles of endotoxin concentration were classified using values for each child who had HE allergy or CM allergy. In children with HE allergy, the values of endotoxin concentration were as follows: Q1, ≤8.0 EU/mg; Q2, >8.0–13.0 EU/mg; Q3, >13.0–23.0 EU/mg; and Q4, >23.0 EU/mg. In children with CM allergy, the values of endotoxin concentration were as follows: Q1, ≤9.3 EU/mg; Q2, >9.3–14.0 EU/mg; Q3, >14.0–20.0 EU/mg; and Q4, >20.0 EU/mg.

### 3.3. Endotoxin Concentrations and the Toleration Statuses of HE and CM

We used univariate analyses to determine the associations between each background factor and tolerance to HE and CM at 2 years old. In children with CM allergy, there was a significant difference between CM tolerance in those with eczema within the past year and those without eczema (Table 1). We conducted a logistic regression analysis of the association between the quartile groups of endotoxin concentration collected from the mattresses of these children at 18 months old and tolerance to HE and CM at 2 years old and adjusted for the presence of eczema as a covariate. Children with HE allergy in their first year of life did not show significant differences in HE tolerance when the Q1 group of endotoxin concentration was compared with the Q2, Q3, and Q4 groups (adjusted odds ratio [aOR]: 1.66, 95% confidence interval (CI): 0.74–3.71; aOR: 1.02, 95% CI: 0.47–2.21; and Q4: aOR: 1.37, 95% CI: 0.61–3.06; respectively). Similarly, children with CM allergy in their first year of life did not show significant differences in CM tolerance when the Q1 group of endotoxin concentration was compared with the Q2, Q3, and Q4 groups (aOR: 1.10, 95% CI: 0.27–4.58; aOR: 1.09, 95% CI: 0.24–5.01; and aOR: 2.68, 95% CI: 0.57–12.63; respectively; Figure 2).

### 3.4. Endotoxin Concentrations and EW- and CM-Specific Ig Levels

At 2 years old, the serum EW- and CM-specific IgE and IgG4 levels, respectively, were measured and compared between children in the Q1 endotoxin concentration group and those in the Q2, Q3, and Q4 groups by Mann–Whitney U test. No significant differences were detected (Figure 3).

### 3.5. EW- and CM-Specific Ig Levels and Toleration Statuses

At 2 years old, the serum EW−specific IgE and IgG4 levels, respectively, were measured and compared between children with tolerance to HE and those in children without tolerance to HE by Mann–Whitney U test. The children with a tolerance of HE had lower HE-specific IgE levels and higher HE-specific IgG4 levels than the children without tolerance of HE (IgE; with tolerance: 95.5 [46.5–231.9] vs. without tolerance: 258.3 [75.2–589.2], *p*−value: <0.001, IgG4; with tolerance: 1965.7 [970.9–3180.5] vs. without tolerance: 335.0 [115.5–1117.9], *p*−value: <0.001). At 2 years old, the serum CM−specific IgE and IgG4 levels, respectively, were measured and compared between children with tolerance to CM and those in children without tolerance to CM by Mann–Whitney U test. The children with CM tolerance had lower CM-specific IgE levels and higher CM-specific IgG4 levels than the children without tolerance of HE (IgE; with tolerance: 69.7 [42.8–174.6] vs. without tolerance: 665.8 [141.7–1974.0], *p*-value: <0.001, IgG4; with tolerance: 311.3 [112.1–676.4] vs. without tolerance: 71.4 [58.3–124.1], *p*-value: <0.001; Figure 4).

### 3.6. Subgroup Analysis by Eczema in the Toleration Statuses of HE and CM

We conducted subgroup analyses of both children with and without eczema to investigate associations between endotoxin concentration and tolerance to HE and CM at 2 years old. These analyses compared children in the Q1 group of endotoxin concentration with those in the Q2, Q3, and Q4 groups by Fisher exact test. In children with HE allergy, the analysis of the subgroup of children with eczema showed a significant difference in HE tolerance between children in the Q1 group and children in the Q2 group (26.1% vs. 64.3%, *p*-value = 0.038; Figure 5). In children with CM allergy, the analysis of the subgroup of children without eczema showed a significant difference in the CM tolerance between children in the Q1 group and children in the Q4 group (50.0% vs. 100.0%, *p*-value = 0.033).

### 3.7. Subgroup Analysis by Eczema in EW- and CM-Specific IgE and IgG4 Levels

We conducted subgroup analyses of both children with and without eczema to investigate associations between endotoxin concentration and allergen-specific IgE and IgG4 levels at 2 years old. These analyses compared children in the Q1 group of endotoxin concentration with those in the Q2, Q3, and Q4 groups by Mann–Whitney U test. There were no significant differences in all EW- and CM-specific IgE levels among the quartile groups of endotoxin concentration (Figure 6 and Figure 7). In children with an EW-specific IgG4 level, analysis of the subgroup of those with eczema revealed that the children in the Q1 group had lower EW-specific IgG4 levels than those in the Q3 group (median [first quartile–third quartile]: Q1: 306.5 [107.4–1575.8] BUg4/mL vs. Q3: 830.0 [264.1–3069.7] BUg4/mL; *p*-value: 0.041; Figure 6). In children with CM allergy, analysis of the subgroup of those without eczema revealed that the children in the Q1 had lower CM-specific IgG4 levels than those in the Q4 group (Q1: 86.8 [61.6–250.1] BUg4/mL vs. Q4: 189.2 [117.8–571.3] BUg4/mL; *p*-value: 0.010; Figure 7).

### 3.8. Correlation between Endotoxin Concentration and Specific IgE and IgG4 Levels

Since it was possible that food allergen-specific IgG4 levels were increased by ingesting foods containing those allergens, we examined the correlation analysis between endotoxin concentration and EW- and CM-specific IgE and IgG4 levels in both children with and without eczema. Only CM-specific IgG4 levels in children without eczema showed a weak but significant correlation with endotoxin concentration (Spearman rank correlation coefficient: 0.35, *p*-value: 0.033; Table 2).

## 4. Discussion

This study did not reveal any association between endotoxin concentration at 18 months old and tolerance to HE and CM at 2 years old in children previously diagnosed with FA against HE and CM. In children with HE allergy and eczema, the percentage of children who had tolerance to HE in the Q2 group of endotoxin concentration was higher than those in the Q1 group. In children with CM allergy without eczema, the percentage of children who had tolerance to CM in the Q4 group of endotoxin concentration was higher than those in the Q1 group. Regarding allergen-specific Ig levels, children with eczema had significantly higher levels of EW-specific IgG4 in the Q3 group of endotoxin concentration than those in the Q1 group. Furthermore, the children with CM allergy without eczema had a higher CM-specific IgG4 level in the Q4 group of endotoxin concentration than those in the Q1 group. Moreover, there was a correlation between CM-specific IgG4 levels and endotoxin concentration.

There were no significant associations between endotoxin concentration and tolerance to causal foods. This may be due to the interaction between endotoxin concentration and the presence of eczema. It is known that endotoxin plays a suppressive role in allergic asthma but is an aggravating factor in whole bronchial allergic asthma [21,38]. Moreover, the presence of eczema negatively affects innate cytokine responses through T cells induced by lipopolysaccharide stimulation in early childhood [39]. These findings indicate that different allergic backgrounds of children may affect the role of endotoxin in the prognosis of FA.

Some analyses of subgroups in this study showed that children in some higher groups of endotoxin concentration had a higher prevalence of tolerance to HE and CM than children in the Q1 group. This might be due to suppressed mast cell activation and the response to allergen-specific IgE antibodies depending on the concentration of endotoxin [40,41]. Our results suggest that the influence of endotoxin might differ according to the presence of eczema in children with FA to HE and CM.

This study showed that children who had higher endotoxin concentration also had higher EW- and CM-specific IgG4 levels than children in the Q1 (lowest) group of endotoxin concentration. Specifically, in children with CM allergy without eczema, those in the Q4 group of endotoxin concentration had higher CM-specific IgG4 levels than those in the Q1 group. These results were consistent with a status of toleration. Oral immunotherapy, which is the treatment for food allergy, increases allergen-specific IgG4 levels and decreases allergen-specific IgE levels [42,43]. Moreover, it has been suggested that allergen-specific IgG4 might be an efficient inhibitor of IgE-dependent reactions due to the structural characteristics of IgG4 [44]. This study showed that children who had tolerance to HE and CM had higher EW- and CM-specific IgG4 levels. Thus, CM-specific-IgG4 might be one of the intermediaries between endotoxin concentration and toleration statuses. Moreover, immunological changes in children with CM allergy might be associated with different results between children with eczema and children without eczema. However, as mentioned above, there might be other mechanisms that mediate the toleration statuses of FA. Therefore, in the results with HE, the association between toleration statutes against HE and endotoxin concentration might not be consistent with the association between EW-specific IgG4 levels and endotoxin concentration.

Significant differences in the influence of endotoxin were shown between children with HE allergy and children with CM allergy. Some different genetic variants in immunological pathways, such as human leukocyte antigen and Toll-like receptor pathways, have been associated with each FA [45,46,47]. Another article showed that genetic background, such as CD14 polymorphisms, also affects the association between endotoxin concentration and developing eczema and FA. [45,48] In this study, different genetic backgrounds might explain the different results, including the association of eczema between HE allergy and CM allergy.

Moreover, because food allergens are also present in the environment, exposure routes, such as oral exposure and percutaneous exposure due to eczema, might differ depending on the type of food allergen [49,50]. These might be associated with the lack of significant difference between Q1 and Q3 in children with HE allergy with eczema. HE proteins are common antigens found in children’s mattresses, and eczema facilitates epicutaneous sensitization of allergens of EW [49,51]. Because food allergen and endotoxin are decreased by washing bedsheets [52], in children with HE allergy with eczema, children in the Q3 group might have a higher amount of EW than children in the Q1 group. The lack of significant difference between the Q1 and Q3 might be due to the balance between the induction effect by endotoxin in the tolerance to hen’s egg and preventing the effect of egg white allergens through epicutaneous sensitization.

### Limitations

First, children with FA to HE and CM were diagnosed by questionnaires, and oral food challenges were not performed. The presence of eczema may delay the timing of the oral food challenge test and make the actual difference in toleration statuses appear smaller, apart from the immunological statuses. Moreover, it might be associated with the results of associations between toleration states of HE and endotoxin concentration. Future studies using data from children with FA diagnosed using oral food challenges are needed. Second, we did not have information about the amounts of food proteins present in the house dust collected from the mattress of the children. It is thought that environmental exposure to more food proteins makes sensitization to the causal food easier [50,53]. Although EW- and CM-specific IgE levels were not associated with endotoxin concentration, the amount of EW allergen in mattress dust might require investigation in a future study.

## 5. Conclusions

Environmental endotoxin might contribute to FA resolution in some children. This study indicates that environmental factors, as well as oral exposure to an allergen, contribute to FA resolution. Future research should consider an individually based approach against endotoxin according to factors such as eczema and causal foods.

## Figures and Tables

**Figure 1 nutrients-15-00968-f001:**
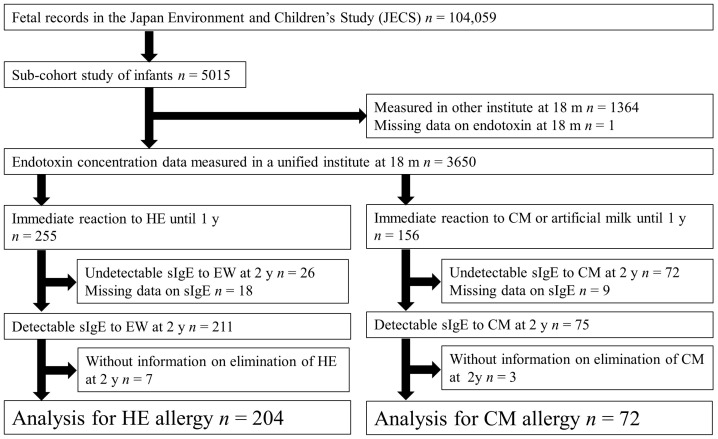
Flowchart of participant inclusion in this study.

**Figure 2 nutrients-15-00968-f002:**
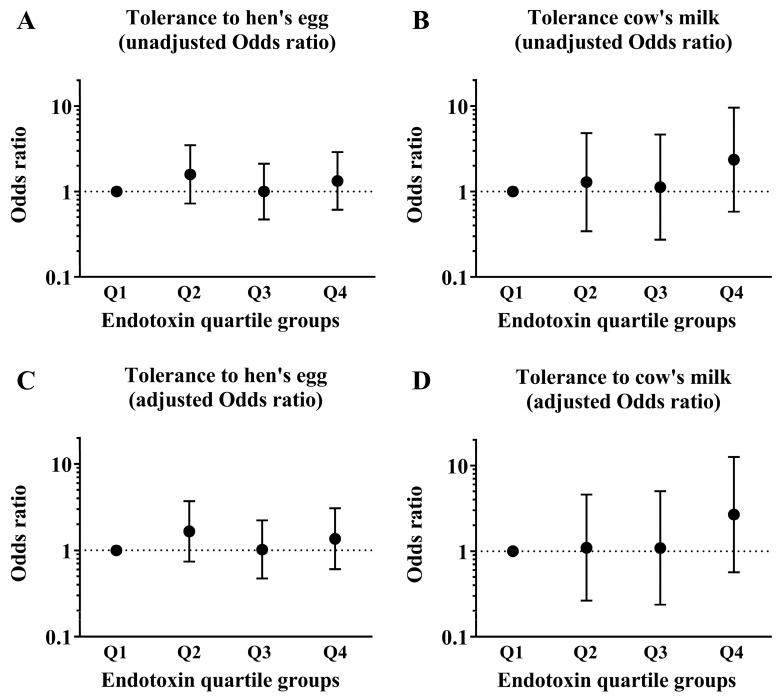
Data of children who had HE allergy and CM allergy in their first year of life. The graphs show the association between the prevalence of tolerance to HE and CM at 2 years old and the quartiles (Q1–Q4) of endotoxin concentrations collected from the mattresses of these children at 18 months old. These show results of (**A**) unadjusted odds ratio in tolerance to HE, (**B**) unadjusted odds ratio in tolerance to CM, (**C**) adjusted odds ratio in tolerance to HE and (**D**) adjusted odds ratio in tolerance to CM.

**Figure 3 nutrients-15-00968-f003:**
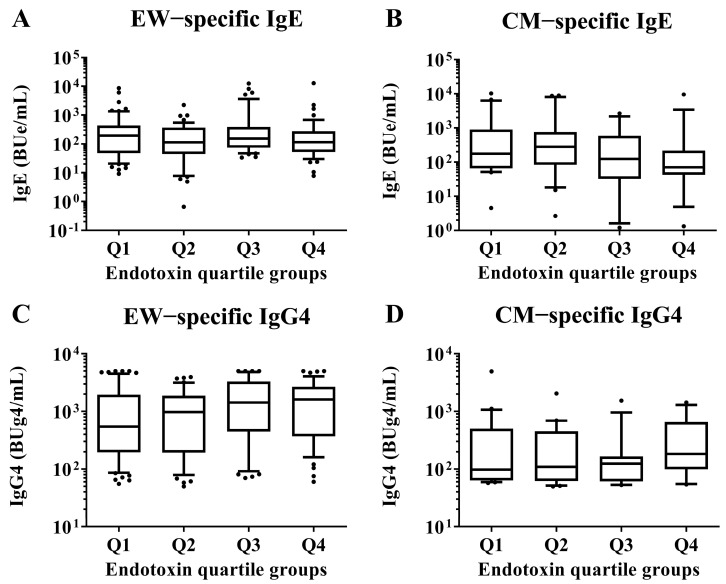
Comparisons of EW- and CM-specific IgE and IgG4 levels in sera of children aged 2 years according to the quartile groups of endotoxin concentration (Q1–Q4) per 1 mg of house dust collected from the mattresses of the children at 18 months old. These show results in (**A**) EW−specific IgE level, (**B**) CM−specific IgE level, (**C**) EW−specific IgG4 level and (**D**) CM−specific IgE level. *p*-values were calculated for each log-converted Ig level using Mann–Whitney U test based on Q1. *p*-values < 0.05 were considered significant. IgE, immunoglobulin E; IgG4, immunoglobulin G4.

**Figure 4 nutrients-15-00968-f004:**
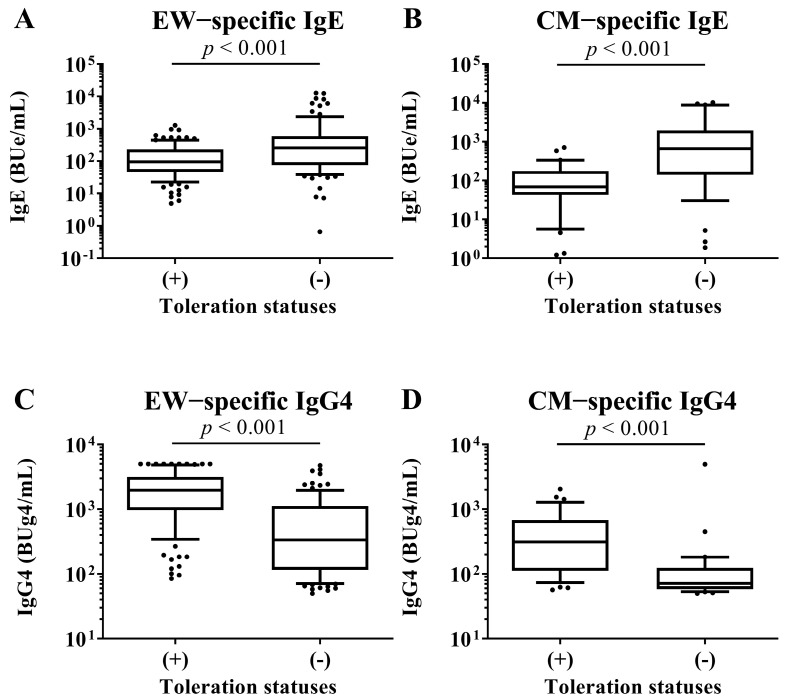
Comparison of EW-specific IgE and IgG4 levels in sera of children at 2 years old between children with tolerance to HE and children without tolerance to HE. Comparison of CM-specific IgE and IgG4 levels in sera of children at 2 years old between children with tolerance to CM and children without tolerance to CM. These show results in (**A**) EW-specific IgE level, (**B**) CM-specific IgE level, (**C**) EW-specific IgG4 level and (**D**) CM-specific IgE level. *p*-values were calculated for each log-converted Ig level using Mann–Whitney U test. *p*-values < 0.05 were considered significant.

**Figure 5 nutrients-15-00968-f005:**
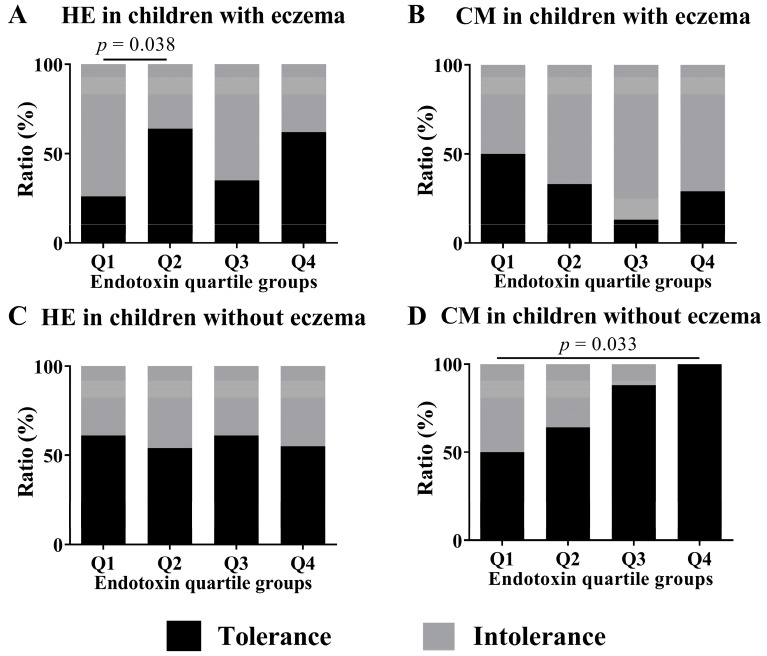
Subgroup analyses by the ratio of tolerance to intolerance at 2 years old in children who had HE and CM allergy in their first year of life and the presence of eczema to assess the association of endotoxin concentrations (Q1–Q4) collected from the mattresses of children at 18 months old. These show results in (**A**) HE in children with eczema, (**B**) CM in children with eczema, (**C**) HE in children without eczema and (**D**) CM in children without eczema. To calculate *p*-values, the values of quartiles were compared using Fisher exact test, with Q1 values as references.

**Figure 6 nutrients-15-00968-f006:**
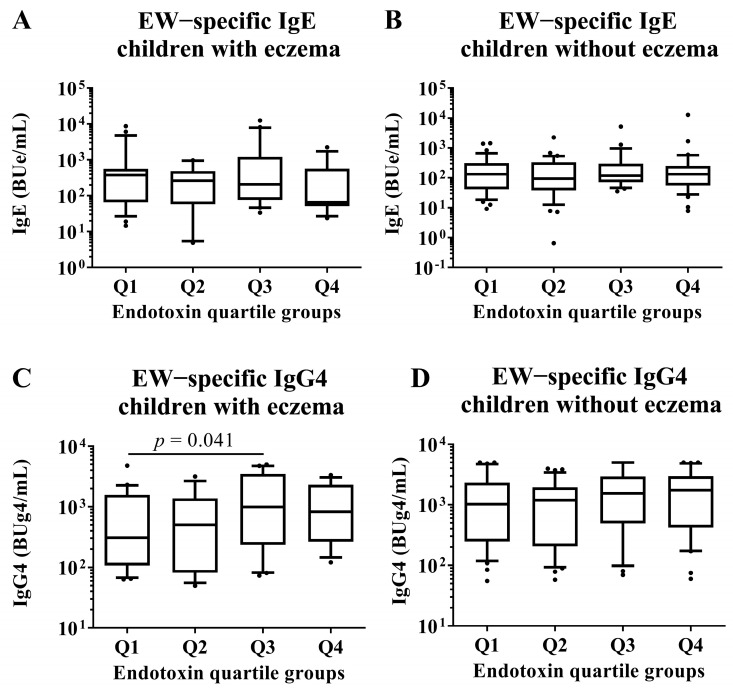
Subgroup analyses by the presence of eczema in children who had HE allergy in their first year of life. Comparison of EW-specific IgE and IgG4 levels in sera of children at 2 years old among quartile groups of endotoxin concentration (Q1–Q4) per 1 g of house dust collected from the mattresses of the children at 18 months old. These show results in (**A**) EW−specific IgE level in children with eczema, (**B**) EW−specific IgE level in children without eczema, (**C**) EW−specific IgG4 level in children with eczema and (**D**) EW−specific IgE level in children without eczema. *p*-values were calculated for each log-converted Ig level using Mann–Whitney U test based on Q1. *p*-values < 0.05 were considered significant.

**Figure 7 nutrients-15-00968-f007:**
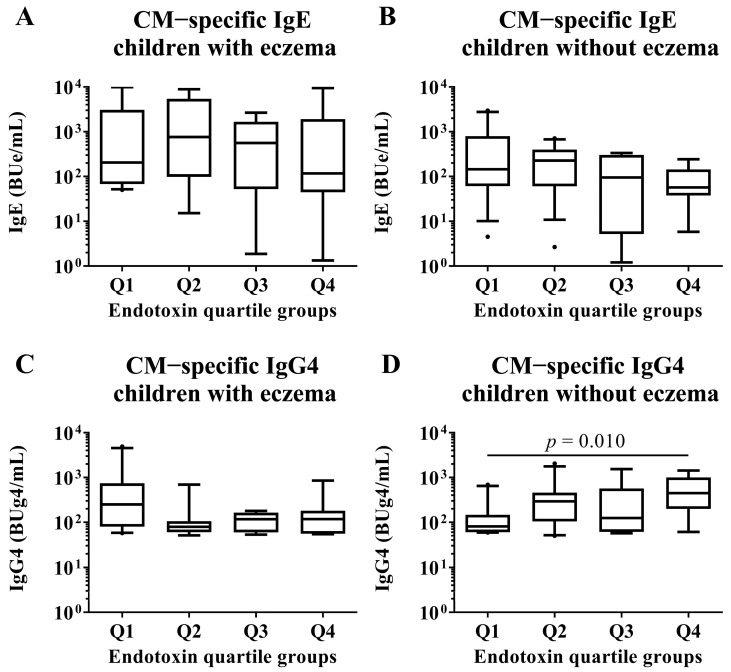
Subgroup analyses by the presence of eczema in children who had CM allergy in their first year of life. Comparisons of CM-specific IgE and IgG4 levels in sera of children at 2 years old among quartile groups of endotoxin concentration (Q1–Q4) per 1 mg of house dust collected from the mattresses of children at 18 months old. These show results in (**A**) CM−specific IgE level in children with eczema, (**B**) CM−specific IgE level in children without eczema, (**C**) CM−specific IgG4 level in children with eczema and (**D**) CM−specific IgE level in children without eczema. *p*-values were calculated for each log-converted Ig level using Mann–Whitney U test based on Q1. *p*-values < 0.05 were considered significant.

**Table 1 nutrients-15-00968-t001:** Characteristics of children in this study.

	Hen’s Egg Allergy at 1 y		Cow’s Milk Allergy at 1 y	
	All, N ^#^	Intake at 2 y N (%) in the Item	All, N ^#^	Intake at 2 y N (%) in the Item
Sex	204		72	
Male	107	53 (49.5)	45	24 (53.3)
Female	97	54 (55.7)	27	14 (51.9)
Mode of delivery	204		72	
Cesarean section	43	18 (41.9)	14	7 (50.0)
Vaginal	161	89 (55.3)	58	31 (53.4)
Birth weight	204		72	
<2500 g	12	9 (75.0)	2	1 (50.0)
≥2500 g	192	98 (51.0)	70	37 (52.9)
Sibling	203		72	
YES	108	54 (50.0)	56	27 (48.2)
NO	95	52 (54.7)	16	11 (68.8)
Maternal allergy	204		72	
YES	105	56 (53.3)	47	23 (48.9)
NO	99	51 (51.5)	25	15 (60.0)
Household Income	201		66	
<4 million yen	74	46 (62.2)	21	14 (66.7)
≥4 million yen	127	60 (47.2)	45	20 (44.4)
Climate of residence	204		72	
Subarctic	49	30 (61.2)	21	14 (66.7)
Temperate	155	77 (49.7)	51	24 (47.1)
Indoor pet owner at 6 m	204		72	
YES	22	15 (68.2)	3	1 (33.3)
NO	182	92 (50.5)	69	37 (53.6)
Breastfeeding until 1 y	204		72	
YES	145	69 (47.6)	54	25 (46.3)
NO	59	38 (64.4)	18	13 (72.2)
Current eczema at 2 y	202		71	
YES	70	30 (42.9)	34	11 (32.4)
NO	132	76 (57.6)	37	27 (73.0)

Abbreviations: 6 m; 6 months of age, 1 y; 1 year of age, 2 y; 2 years of age. ^#^: Number of children without missing value.

**Table 2 nutrients-15-00968-t002:** Correlation between endotoxin concentration and EW- and CM-specific IgE and IgG4 levels in subgroups of children with and without eczema.

Specific Immunoglobulin	Allergen	Current Eczema	Correlation Coefficient	*p*-Value
Specific IgE	Egg white	Yes	−0.087	0.48
		No	0.014	0.87
	Cow’s milk	Yes	−0.11	0.54
		No	−0.28	0.090
Specific IgG4	Egg white	Yes	0.24	0.048
		No	0.18	0.042
	Cow’s milk	YES	−0.16	0.36
		No	**0.35**	**0.033**

Correlation coefficients and *p*-values were calculated by Spearman’s rank correlation coefficient. When the correlation coefficient ≥0.30 and *p*-value < 0.05 were satisfied, the items were defined as significant. Bold numbers mean significant items.

## Data Availability

Data are unsuitable for public deposition due to ethical restrictions and the legal framework of Japan. It is prohibited by the Act on the Protection of Personal Information (Act No. 57 of 30 May 2003, amendment on 9 September 2015) to publicly deposit data containing personal information. Ethical Guidelines for Medical and Health Research Involving Human Subjects enforced by the Japan Ministry of Education, Culture, Sports, Science and Technology and the Ministry of Health, Labour and Welfare also restrict the open sharing of epidemiologic data. All inquiries about access to the data should be sent to: jecs-en@nies.go.jp. The person responsible for handling inquiries sent to this e-mail address is Shoji F. Nakayama, JECS Programme Office, National Institute for Environmental Studies.

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
