# Peer review of "Relationship between Food Allergy and Endotoxin Concentration and the Toleration Status at 2 Years: The Japan Environment and Children’s Study"

_nutrients, 2023, doi:10.3390/nu15040968_

Round 1

Reviewer 1 Report

The authors present their manuscript looking at children with hen's egg or milk allergy and concentration of endotoxin in their environment.  Only is a subgroup analysis did the authors find any significant results. 

Major weakness of the study is the definitions of disease are survey and patient reported.  Also the authors too strongly rely on the implications of poorly defined biomarkers.  While it is true specific IgG4 increases with oral immunotherapy, there is no evidence that patients with higher IgG4 because of endotoxin are undergoing the same process.  The statement in Line 292-293 is not supported.  Further if true, they should show in that group if possible that they develop tolerance sooner than their peers. 

Is the endotoxin level on average of the patients with food allergy different than the 100,000ish patients without food allergy?

Line 37-38 The incidence of naturally resolving hen's egg or cow's milk allergy is NOT 50%.  It is significantly greater than 50%.  Cow's milk allergy resolves in ~80-85% of children while hen's egg is lower but generally thought to be ~70-75%.  In fact, it is generally thought that egg and milk allergy are easy to achieve natural tolerance.  Citations provided may look only at resolution by age 2, but as stated this is incorrect. 

Author Response

Dear Editor:

We thank you and the referees for reviewing our manuscript and giving us important suggestions. We have revised our manuscript based on the comments. Our point-by-point responses are provided below, and the revisions are highlighted. We hope to resolve the concerns with this modification.

#Reviewer1

Major weakness of the study is the definitions of disease are survey and patient reported.  Also the authors too strongly rely on the implications of poorly defined biomarkers.  While it is true specific IgG4 increases with oral immunotherapy, there is no evidence that patients with higher IgG4 because of endotoxin are undergoing the same process.  The statement in Line 292-293 is not supported. Further if true, they should show in that group if possible that they develop tolerance sooner than their peers.

Response: Thank you for the comments. Yes, we demonstrated the limitation of the outcome assessment in the discussion (lines 348-349). We compared egg white (EW) and cow’s milk (CM)-specific IgE and IgG4 levels when the children were 2 years old in toleration statuses of hen’s egg (HE) and CM. The children with the tolerance of (HE) had lower HE-specific IgE levels and higher HE-specific IgG4 levels than the children without tolerance of HE, and the children with the tolerance of CM had lower CM-specific IgE levels and higher CM-specific IgG4 levels than the children without tolerance of HE (We added them to sections of methods (Line 141-144) and results (Line 206-225) and made figure (Figure4)). However, as the reviewer 1 pointed out, we understand that we could not consider it equal to mechanisms in oral immunotherapy, so we cited the other article and changed it to state that IgG4 may be one of the intermediaries between endotoxin concentration and toleration statuses. (Line320-332)

Lines 141–144: Moreover, to confirm of association between EW- and CM-specific immunoglobulin levels and status of toleration statuses, we used Mann–Whitney U tests to compare EW- and CM-specific IgE and IgG4 levels when the children were 2 years old in toleration statuses of HE and CM. 

Lines 206–225: EW- and CM-specific Ig levels and toleration statuses

At 2 years old, the serum EW-specific IgE and IgG4 levels, respectively, were measured and compared between children with tolerance to HE and those in children without tolerance to HE by Mann–Whitney U test. The children with the tolerance of HE had lower HE-specific IgE levels and higher HE-specific IgG4 level than the children without tolerance of HE (IgE; with tolerance: 95.5 [46.5–231.9] vs. without tolerance: 258.3 [75.2–589.2], P-value: <0.001, IgG4; with tolerance: 1965.7 [970.9–3180.5] vs. without tolerance: 335.0 [115.5–1117.9], P-value: <0.001). At 2 years old, the serum CM-specific IgE and IgG4 levels, respectively, were measured and compared between children with tolerance to CM and those in children without tolerance to CM by Mann–Whitney U test. The children with the tolerance of CM had lower CM-specific IgE levels and higher CM-specific IgG4 levels than the children without tolerance of HE (IgE; with tolerance: 69.7 [42.8–174.6] vs. without tolerance: 665.8 [141.7–1974.0], P-value: <0.001, IgG4; with tolerance: 311.3 [112.1–676.4] vs. without tolerance: 71.4 [58.3–124.1], P-value: <0.001) (Figure 4). 

Figure 4. Comparison of EW-specific IgE and IgG4 levels in sera of children at 2 years old between children with tolerance to HE and children without tolerance to HE. And the comparison of CM-specific IgE and IgG4 levels in sera of children at 2 years old between children with tolerance to CM and children without tolerance to CM. P-values were calculated for each log-converted Ig level using Mann–Whitney U test. P-values <0.05 were considered significant.

Lines 320–332: Oral immunotherapy, which the treatment for food allergy, increases allergen-specific IgG4 levels and decreases allergen-specific IgE levels [42,43]. Moreover, it has been suggested that allergen-specific IgG4 might be an efficient inhibitor of IgE-dependent reactions due to the structural characteristics of IgG4. This study showed that children who had tolerance to HE and CM had higher EW- and CM-specific IgG4 levels. Thus, CM-specific-IgG4 might be one of the intermediaries between endotoxin concentration and toleration statuses. Moreover, immunological changes in children with CM allergy might be associated with different results between children with eczema and children without eczema. However, as mentioned above, there might be other mechanisms that mediate the toleration statuses of FA. Therefore, in the results with HE, the association between toleration statutes against HE and endotoxin concentration might not be con-sistent with the association between EW-specific IgG4 levels and endotoxin concentra-tion.t ent with the association between EW-specific IgG4 levels and endotoxin concentration.

Is the endotoxin level on average of the patients with food allergy different than the 100,000ish patients without food allergy?

Response: Since we measured the endotoxin concentrations in children, including the sub-cohort study, as we mentioned in the method section (lines 62-67), we do not have data on 100.000 children with endotoxin concentrations.

Line 37-38 The incidence of naturally resolving hen's egg or cow's milk allergy is NOT 50%.  It is significantly greater than 50%. Cow's milk allergy resolves in ~80-85% of children while hen's egg is lower but generally thought to be ~70-75%.  In fact, it is generally thought that egg and milk allergy are easy to achieve natural tolerance.  Citations provided may look only at resolution by age 2, but as stated this is incorrect.

Response: We understand the concern. We cited other studies that followed the natural history of food allergy until puberty (references 8-11). We have added to sentences that many children with food allergy to hen's egg and cow's milk could archive tolerance against those by puberty. Moreover, we have added the words "by 2 years of age" to our previous texts (Line 40).

Lines 37–43: Although several studies showed that 68-94% of children with FA against HE and more than 90% of children with FA against CM acquired tolerance after puberty, other articles showed only 50% of cases of children with FA against HE and CM resolve naturally by 2 years of age. In a part of children, it is difficult to archive early natural tolerance to HE and CM.

Reviewer 2 Report

In this study by Irahara et al, the authors analyze the relationship between endotoxin levels in households and the likelihood of reaching tolerance in food allergic individuals at the age of two years. It is very possible that reliance on questionnaires and not oral food challenges may have led to misdiagnoses and obscure potential results of this study.  The authors are able to demonstrate that by subgroup analysis separated by the presence or absence of eczema that the development of tolerance to HE and CM can be correlated to the endotoxin levels in the children with eczema (HE) and children without eczema (CM), respectively.  

1.    Please describe for the reader what were symptoms reported (line 80) in the food allergic cohorts? A summary chart in the online supplemental would be useful.

2.    Figure 4-Why do the authors think in HE with eczema there was no significant difference between Q1 and Q3?

3.    Perhaps the authors can give a rationale as to why the ratio of tolerance increases in CM-allergic patients without eczema who are exposed to higher levels of endotoxin but not in children with eczema.

4.    If the ratio of tolerance increases between Q1 and Q2 endotoxin quartile groups, why is the EW-specific igG4 different between Q1 and Q3 in children with eczema and not seen in Q2.

Author Response

Dear Editor:

We thank you and the referees for reviewing our manuscript and giving us important suggestions. We have revised our manuscript based on the comments. Our point-by-point responses are provided below, and the revisions are highlighted. We hope to resolve the concerns with this modification.

Reviewer: 2

  1. Please describe for the reader what were symptoms reported in the food allergic cohorts? A summary chart in the online supplemental would be useful.

Response: We have made the table describing symptoms in children with hen’s egg and cow’s milk allergies and added them as supplementary material (Table S1). Moreover, we have added to the results sections the number of children with itchy skin and rashes, which was the most frequent symptom (Lines 158-160).

Lines 158–160: Of them, 170 children (83.3%) had itchy skin and rashes in children with HE allergy, and 56 children (77.8%) had these symptoms in children with CM allergy (Table S1).

  1. Figure 4-Why do the authors think in HE with eczema there was no significant difference between Q1 and Q3?

Response: We think it is a result due to the balance between the induction effect by endotoxin in the tolerance to hen's egg and preventing the effect of egg white allergens in it through epicutaneous sensitization. Allergens of egg white detected in many children's bedsheets, and eczema facilitates epicutaneous sensitization of allergens of egg white. Washing sheets may decrease food allergen and endotoxin. Therefore, children in the Q3 group may have a higher amount of allergen of egg white than children in the Q1 group. We consider that the lack of significant difference between Q1 and Q3 resulted from those effects being antagonistic. We have added these considerations to the discussion section (Lines 346-354).

Lines 346–354: These might be associated with the lack of significant difference between Q1 and Q3 in children with HE allergy with eczema. HE proteins are common antigens found in children's mattresses, and eczema facilitates epicutaneous sensitization of allergens of EW [49, 51]. Because food allergen and endotoxin are decreased by washing bedsheets [52], in children with HE allergy with eczema, children in the Q3 group might have a higher amount of EW than children in the Q1 group. The lack of significant difference between the Q1 and Q3 might be due to the balance between the induction effect by endotoxin in the tolerance to hen's egg and preventing the effect of egg white allergens through epicutaneous sensitization.

  1. Perhaps the authors can give a rationale as to why the ratio of tolerance increases in CM-allergic patients without eczema who are exposed to higher levels of endotoxin but not in children with eczema.

Response: In children without eczema, endotoxin concentrations were positively correlated with cow’s milk-specific IgG4 levels. Another study showed IgG might inhibit IgE-dependent reactions due to the structural characteristics of IgG4. However, in children with eczema, endotoxin concentrations were not correlated with milk-specific IG4 levels. One possibility is that the association between endotoxin concentration and cow’s milk specific-IgG4 levels affects the different mechanisms in children with and without eczema. We have added these to the discussion section (Lines 322–328). In addition, genetic factors may be associated with endotoxin concentration, eczema, and toleration statuses of cow’s milk. CD14 polymorphisms have been suggested for the association between endotoxin concentration and eczema. CD14 polymorphisms have also been associated with the presence of food allergy. In this study, different genetic backgrounds might explain the different results between children with and without eczema. We have added to the discussion sections (Lines 339–343).

Lines 322–332: Moreover, it has been suggested that allergen-specific IgG4 might be an efficient inhibitor of IgE-dependent reactions due to the structural characteristics of IgG4. This study showed that children who had tolerance to HE and CM had higher EW- and CM-specific IgG4 levels. Thus, CM-specific-IgG4 might be one of the intermediaries between endotoxin concentration and toleration statuses. Moreover, immunological changes in children with CM allergy might be associated with different results between children with eczema and children without eczema. However, as mentioned above, there might be other mechanisms that mediate the toleration statuses of FA. Therefore, in the results with HE, the association between toleration statutes against HE and endotoxin concentration might not be consistent with the association between EW-specific IgG4 levels and endotoxin concentration.

Lines 339–343: Another article showed that genetic background, such as CD14 polymorphisms, affects the association between endotoxin concentration and developing eczema and FA. In this study, different genetic backgrounds might explain the different results, including the association of eczema between HE allergy and CM allergy.

  1. If the ratio of tolerance increases between Q1 and Q2 endotoxin quartile groups, why is the EW-specific IgG4 different between Q1 and Q3 in children with eczema and not seen in Q2

Response: As noted in the text, endotoxin is also known to affect the activation of mast cells. This point suggests that mechanisms other than the elevation of food allergen-specific IgG4 levels may also affect toleration statuses in food allergy. Therefore, the association toleration statutes against hen's egg and endotoxin concentration might not be consistent with the association between EW-specific IgG4 levels and endotoxin concentration. We have added these sentences to the discussion section (Lines 328–332).

Furthermore, the presence of eczema may delay the timing of the oral food challenge test. It may make the actual difference in toleration statuses appear smaller, apart from the immunological statutes. Moreover, it might be associated with the results of associations between toleration states of hens' eggs and endotoxin concentration. This point is one of the limitations of this study. So, we have added these to the limitation section. Furthermore, we have described that future studies using data from children with food allergy diagnosed by oral food challenges are needed. (Lines 358–362).

Lines 328–332: However, as mentioned above, there might be other mechanisms that mediate the toleration statuses of FA. Therefore, in the results with HE, the association between toleration statutes against HE and endotoxin concentration might not be consistent with the association between EW-specific IgG4 levels and endotoxin concentration.

Lines 358–363: The presence of eczema may delay the timing of the oral food challenge test and make the actual difference in toleration statuses appear smaller, apart from the immunological statuses. Moreover, it might be associated with the results of associations between toleration states of HE and endotoxin concentration. Future studies using data from children with FA diagnosed by oral food challenges are needed.

Round 2

Reviewer 2 Report

Q1. The authors in the revised version removed the sentence stating, "This study included children with an immediate allergic reaction…line 64 in the original. Why? Are they reporting on immediate reactions or not?

Q2. They provided in Table S1 the symptoms of the patients in the study.  Diarrhea is not a symptom of an immediate IgE-mediated reaction. If this is the only symptom in some patients and they are reporting on immediate reactions, these patients should be removed from the study. In any event, diarrhea is not an IgE-mediated  reaction seen

Q3. The scientific term for itchiness during an allergic reaction is pruritis. Please change to that term.

Q4. What is meant by rashes? Erythema ? urticaria? If so then these terms should be used.

Q5. The term convulsion can be removed from the table, since none had this symptom.

Author Response

Dear Editor:

We thank the referees for reviewing our manuscript and giving valuable suggestions. We have revised our manuscript based on the comments. Our point-by-point responses are provided below, and the revisions are highlighted. We hope to resolve the concerns with this modification.

Reviewer: 2

Q1. The authors in the revised version removed the sentence stating, "This study included children with an immediate allergic reaction…line 64 in the original. Why? Are they reporting on immediate reactions or not?

Response: We moved the sentence that reviewer 2 pointed out "This study included children with an immediate allergic reaction after ingestion of HE or CM during their first year of life and endotoxin concentration data measured in a unified institute when the children were 18 months old" to line 67 in the revised version.

Q2. They provided in Table S1 the symptoms of the patients in the study. Diarrhea is not a symptom of an immediate IgE-mediated reaction. If this is the only symptom in some patients and they are reporting on immediate reactions, these patients should be removed from the study. In any event, diarrhea is not an IgE-mediated reaction seen

Response: We understand the concern. However, another article described some children with immediate food allergy have diarrhea through IgE-mediated mechanisms (Gargano, D.; Appanna, R.; Santonicola, A.; De Bartolomeis, F.; Stellato, C.; Cianferoni, A.; Casolaro, V.; Iovino, P. Food Allergy and Intolerance: A Narrative Review on Nutritional Concerns. Nutrients 2021, 13, doi:10.3390/nu13051638.). We selected children with detectable egg white- and cow’s milk-specific IgE and immediate reaction within 3 hours from intake of foods in this study. We hope reviewer 2 approves including those children in this study.

Q3. The scientific term for itchiness during an allergic reaction is pruritis. Please change to that term.

Response: Thank you for the suggestion. We have changed "itchy skin" to "pruritus." (Lines 158 in main text and table S1)

Q4. What is meant by rashes? Erythema ? urticaria? If so then these terms should be used.

Response: Thank you for the suggestion. e have fixed this word to "urticaria."(Lines 158 in main text and table S1)

Q5. The term convulsion can be removed from the table since none had this symptom.

Response: Thank you for the suggestion. We have deleted the line of convulsion from the table. (Table S1)
